# Spatial Overlay Analysis of Geochemical Singularity Index α-Value of Porphyry Cu Deposit in Gangdese Metallogenic Belt, Tibet, Western China

**Shunli Zheng [1], Xiaojia Jiang [2,*] and Shunbao Gao [3]**

[1]  School of Earth Resources, China University of Geosciences, Wuhan 430074, China; shunlizheng@163.com
[2]  Collaborative Innovation Center for Exploration of Strategic Mineral Resources, School of Earth Resources, China University of Geosciences, Wuhan 430074, China
[3]  Institute of Geological Survey, China University of Geosciences, Wuhan 430074, China; gaoshunbao2002@163.com
[*]  Correspondence: jiangxiaojia@cug.edu.cn

**Abstract:** The statistical modeling with ILR-RPCA-back CLR has two problems when dealing with the closure effect of geochemical data. Firstly, after performing isometric logratio (*ilr*) transformation, robust principal component analysis (RPCA) is employed for processing. The double-plot diagram illustrates that the element sequence transformation occurs in the first and second principal components, while the unique principal component remains unattainable. Secondly, by transforming both the score and load into the centered logratio (CLR) space using the U matrix, it is possible to obtain a score result that corresponds to the original order of elements according to the CLR = ILR·U formula. However, for obtaining a load result that corresponds to the original order of elements, an alternative formula "CLR = UT·ILR" must be used instead. In order to determine the optimal element assemblage for porphyry copper deposits, this study conducted statistical analysis on mineral assemblages from discovered deposits in the Gangdese metallogenic belt and identified Cu, Mo, Au, Ag, W, and Bi as key elements associated with porphyry copper deposits. Subsequently, by analyzing the singularities of the composite elements, the spatial overlay of the combined element is carried out, and concentration-area (C-A) fractal filtering is applied to identify the anomaly and background areas. To facilitate comparison, we conducted an analysis of various mineral and ore deposit types, revealing the following findings: (1) Combination elements exhibit superior recognition capability than single elements in porphyry copper deposits; (2) Skarn-type copper deposits unrelated to porphyry show a high degree of dissimilarity compared to those related to porphyry; (3) this method offers advantages over the single element method in evaluating porphyry gold deposits by reducing anomaly levels and initial investment during the evaluation stage for porphyry copper anomalies; (4) However, this method has limited ability in distinguishing between porphyry copper and molybdenum deposits.

**Keywords:** geochemical singularity index; spatial overlay analysis; Gangdese metallogenic belt; porphyry Cu deposit

## 1. Introduction

Porphyry deposits are globally significant sources of copper, molybdenum, and gold. Among them, porphyry copper deposits contribute approximately 55% of the world's copper reserves, while porphyry molybdenum deposits account for approximately 90% of global molybdenum reserves. Additionally, porphyry gold deposits make up around 20% of the world's gold reserves. Notably, China possesses a significant proportion of porphyry deposits [1–4]. In recent years, extensive exploration and research efforts have been conducted on large-scale porphyry copper mines such as Qulong, Jiama, Xiongcun, Duolong, and Tiegelongnan. These endeavors have revealed the Qinghai-Tibet Plateau as a region of utmost importance and potential for copper production in China [5].

Mineralization is a process that leads to the concentration and accumulation of valuable materials within a relatively limited temporal and spatial range, as compared to normal regional geological processes [6]. The complexity of the metallogenic process stems from the intricate dynamics of multi-component and multi-geological interactions, coupled with overlay and other complex processes [7]. Relying solely on the exploration of single elements through geochemical analysis is insufficient for the porphyry copper mine prospecting work. Therefore, the identification of combination elements plays a crucial role in geochemical data processing. Currently, methods such as principal component analysis (PCA), robust principal component analysis (RPCA), factor analysis (FA), and robust factor analysis (RFA) are employed for combination element identification [8–13].

These methods often require logarithmic transformations or logratio transformations (such as alr: additive logratio, clr: centered logratio, *ilr*: isometric logratio, etc.) to eliminate the closure effect of compositional data. However, they encounter challenges during the analysis process, such as difficulties in interpretation and singularity effects, which are difficult to resolve. To address these issues, some researchers have proposed the ILR-RPCA-back CLR statistical model. This model involves transforming the raw geochemical data through an isometric logarithm (ILR) transformation and subsequently applying robust principal component analysis (RPCA) for data processing. The scores and loads obtained from RPCA are then transformed into the centered logratio (CLR) space using the U-matrix [14]. While this method has been applied in the analysis of geochemical data related to magma-hydrothermal ore deposits [14,15], challenges in interpretation and other aspects persist (please refer to Section 4 for more details). Moreover, the combination elements extracted during the analysis of the above method, particularly in small-scale geochemical data, are predominantly associated with environmental (background) elements rather than ore-forming elements. In the Gangdese metallogenic belt, the porphyry copper deposits exhibit a distinct combination of mineralization elements. Utilizing spatial overlay analysis pertaining to deposit genesis is more advantageous for identifying porphyry copper deposits.

In the eastern section of the Gangdese metallogenic belt, which is known as the largest district for dense porphyry copper deposits, numerous large and super large deposits like Qulong, Tinggong, Chongjiang, Jiama, and Bairong have been discovered. However, in the western section of the belt, the number of discovered deposits remains relatively limited. Due to the limitations of traditional methods in identifying complex anomalies and low-delay anomalies, it becomes necessary to employ advanced theories and techniques to guide prospecting efforts for porphyry copper deposits in the western section of the Gangdese metallogenic belt. Local singularity analysis is not only a powerful multifractal tool to identify weak anomalies but also a type of local neighborhood statistical analysis that can reduce the effects of regional background and provide useful statistical information by involving the data within a small singularity around a specific spatial location [16–23]. Therefore, local singularity analysis is widely used. This study specifically concentrates on porphyry copper deposits in the western section of the Gangdese metallogenic belt and aims to address the limitations of the *ilr-RPCA-back-clr* model. To investigate the spatial distribution pattern of geochemical elements and identify real anomalies related to porphyry copper deposits, the study employs overlay analysis of the local singular elements of geochemical combination elements.

## 2. Geological Setting and Geochemical Data

### 2.1. Regional Geological Background

The western part of the Gangdese belt is located in the south-central part of Tibet. The tectonic structure belongs to the Gangdese-Nyainqentanglha landmass in the Gangdese-Himalayan tectonic area, which is between the Yarlung Zangbo River and the Bangong Lake-Nujiang Belt [24]. The basement consists of the Nyainqentanglha group from Presinian. The Cambrian and low-middle Triassic strata are generally absent, with more complete development observed in the Ordovician stratigraphy. The Ordovician-Permian strata are

mainly distributed in the northern part of the study area, and the Upper Triassic-Jurassic strata are distributed in the northern part of the study area, while the Cretaceous-Neogene strata are distributed in the whole area (Figure 1). The region exhibits a significant presence of magmatic rocks, characterized by extensive chronological distribution, large exposed areas, wide distribution ranges, complex origins, and diverse types. The intrusive and volcanic rocks have been extensively developed since the Triassic, mainly formed in the Cretaceous-Paleogene, with the latest volcanic magma activity in the Miocene. Pan et al. [25] classified the tectonic unit of the Gangdese orogenic belt. By comparing these tectonic units, the study area can be divided into four magmatic belts consisting of intermediate-acidic magmatic rocks, including South Gangdese, Longgeer-Nyainqentanglha, Ze long, and Ban-Bengcuo. These four magmatic belts are closely associated with the distribution of the porphyry copper-molybdenum-gold deposits. The porphyry copper polymetallic deposits in the study area are mainly distributed in the Gangdese magmatic arc, including Zhunuo, Jiru, Xiongcun, Dongga, etc. Additionally, deposits can be found in the uplifted area of the Longgeer-Nanmulin arc-back belt, which includes Ria in Cuoqin, as well as in the island arc belt of Cuoqin-Xainza, consisting of Balazha, Gaerqiong, and others. The detailed chronology and tectonic setting of the typical porphyry deposits mentioned above have been studied by many authors. The results suggest that the formation environment and metallogenic characteristics of the porphyry deposits in this area are formed in three tectonic settings: island arcs, syn-collisional environments, and post-collisional extensional environments, occurring during four metallogenic periods: the Early Middle Jurassic, Late Cretaceous, Eocene, and Miocen. The porphyry deposits in island arcs such as Dongga and Xiongcun are only distributed in the south of the southern margin of the Gangdese magmatic arc zone. The porphyry deposits in syn-collisional and post-collisional extensional environments such as Zhunuo and Jiru are mainly distributed in the southern Gangdese magmatic arc. Additionally, a few porphyry deposits, such as Balazha and Gaerqiong, are distributed in the northern part of the Cuoqin-Xainza Island arc and the Bangor magmatic arc belt. All these insights greatly extend the geochemical prospecting of porphyry copper polymetallic ore in the western Gangdese.

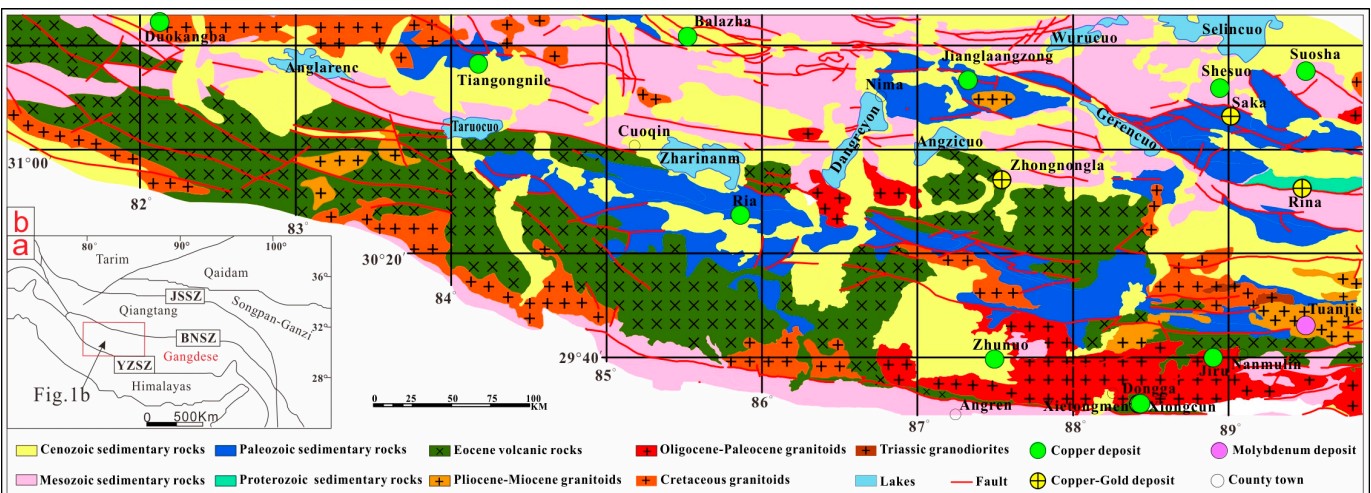

**Figure 1.** (**a**) Major tectonic units and boundary map of the Tibetan Plateau. (**b**) Regional geological map of the western Gangdese.

*2.2. Geochemical Data*

The datasets used in this study were a digital 1:500,000 geological map and a regional stream sediment geochemical dataset (Figure 2). The regional stream sediment geochemical dataset is based on 9546 stream sediment samples collected and analyzed for 39 major, minor, trace, and subtrace elements at a density of one sample per 20 km$^2$ during the Chinese National Geochemical Mapping (CNGM) project as part of the "Regional Geochemistry

National Reconnaissance (RGNR) Project". These 39 elements in stream sediments are determined based on inductively coupled plasma-mass spectrometry (ICP-MS), X-ray fluorescence (XRF), and inductively coupled plasma-atomic emission spectrometry (ICP-AES) as the backbone combined with other methods. The concentration values of Bi, Cd, Co, Cu, La, Mo, Nb, Pb, Th, U, and W are determined by ICP-MS. The concentration values of Al, Cr, Fe, K, P, Si, Ti, Y, and Zr were determined by XRF. The concentration values of Ba, Be, Ca, Li, Mg, Mn, Na, Ni, Sr, V, and Zn were determined by ICP-AES. The concentration values of Ag, B, and Sn were determined by emission spectrometry (ES). The concentration values of As and Sb were determined by hydride generation-atomic fluorescence spectrometry (HG-AFS). The concentration values of Au, Hg, and F are determined by graphite furnace-atomic absorption spectrometry (GF-AAS), cold vapor-atomic fluorescence spectrometry (CV-AFS), and ion-selective electrode [26], respectively. The detection limit for each of the 39 elements is given in Table 1 [27].

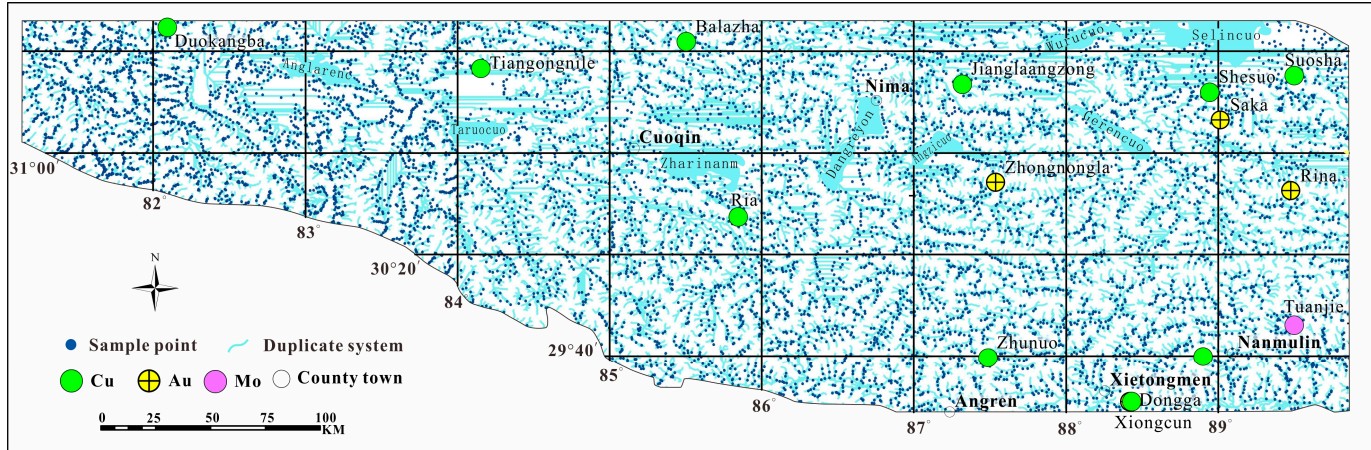

**Figure 2.** Stream sediment sample location map of the western Gangdese.

**Table 1.** The detection limits of 39 elements [28].

| No. | Elements | Detection Limit | No. | Elements | Detection Limit | No. | Elements | Detection Limit |
|-----|----------|-----------------|-----|----------|-----------------|-----|----------|-----------------|
| 1 | Ag | 0.02 | 14 | La | 30 | 27 | U | 0.5 |
| 2 | As | 1 | 15 | Li | 5 | 28 | V | 20 |
| 3 | Au | 0.0003 | 16 | Mn | 30 | 29 | W | 0.5 |
| 4 | B | 5 | 17 | Mo | 0.4 | 30 | Y | 5 |
| 5 | Ba | 50 | 18 | Nb | 5 | 31 | Zn | 10 |
| 6 | Be | 0.5 | 19 | Ni | 2 | 32 | Zr | 10 |
| 7 | Bi | 0.1 | 20 | P | 100 | 33 | $SiO_2$ | 0.10% |
| 8 | Cd | 0.05 | 21 | Pb | 2 | 34 | $Al_2O_3$ | 0.10% |
| 9 | Co | 1 | 22 | Sb | 0.1 | 35 | $TFe_2O_3$ | 0.05% |
| 10 | Cr | 15 | 23 | Sn | 1 | 36 | MgO | 0.05% |
| 11 | Cu | 1 | 24 | Sr | 5 | 37 | CaO | 0.05% |
| 12 | F | 100 | 25 | Th | 4 | 38 | $Na_2O$ | 0.05% |
| 13 | Hg | 0.0005 | 26 | Ti | 100 | 39 | $K_2O$ | 0.05% |

### 3. Methods

*3.1. Multifractal Inverse Distance Weighted (MIDW)*

Kriging and inverse distance weighting (IDW) are two frequently used methods for constructing contours and surfaces to determine spatial variations and anomalies. However, one limitation of Kriging is that it tends to smooth the data due to its reliance on variograms and moving weighted averages [29]. As a result, it may not effectively capture outliers during interpolation. A common drawback of moving average interpolation techniques lies in their failure to consider the local properties of the data. In contrast, the Modified Inverse

Distance Weighting (MIDW) method incorporates a local singularity into the basic model of moving average interpolation, considering spatial correlations such as Kriging [30]. Compared to the traditional interpolation methods, the MIDW method has two significant advantages: (1) it improves the accuracy of interpolation results, and (2) it retains the local structure of the interpolation surface. The MIDW method has been proven to be effective in both exploration and environmental geochemistry [27,30,31].

### 3.2. Local Singularity Spatial Overlay Analysis (α-Value)

The local singularity analysis is another important progress for fractal/multifractal modeling of geochemical data [32]. It is defined as the characterization of the anomalous behaviors of singular physical processes that often result in anomalous amounts of energy release or material accumulation within a narrow spatial–temporal interval [16]. The singularity can be estimated from the observed element concentration within small neighborhoods based on the following equation:

$$X = c \cdot r^{\alpha - E}, \tag{1}$$

where $X$ represents element concentrations, $c$ is a constant value, $\alpha$ is the singularity, $E$ is a normalized distance measure, such as block cell edge, and $E$ is the Euclidian dimension [19].

The window-based method can be used to estimate the singularity index $\alpha$ [16]. Based on either raw point geochemical data or raster maps using the following steps: (1) defining a set of sliding square (or other shapes) windows $A(r)$ with variable window sizes, $r_{min} = < r_1 < r_2 < \ldots < r_n = r_{max}$, and calculating the average concentration $C[A(r_i)]$ for each window size for a given location on the map, where $C[A(r_i)]$ is equal to the sum of all the cell concentrations divided by the total number of cells within the window; (2) plotting the pair data set of $C[A(r_i)]$ ($i = 1, \ldots, n$) and $r_i$ in a log–log graph, the following linear relationship can be obtained.

$$logC[A(r_i)] = c + (\alpha - 2)log(r), \tag{2}$$

The value of $\alpha - 2$ can be estimated from the slope of the fitted straight line, and in step (3), repeat the same procedure to all other locations on the geochemical map. For a geochemical map, most of the areas are linked with a singularity value close to 2, representing a normal distribution, whereas the areas with $\alpha < 2$ or $\alpha > 2$ represent enrichment and depletion of element concentrations, respectively [16].

The method of spatial overlay local singularity analysis proposed in this paper is as follows. First step: determine the effective combination of geochemical elements, perform the local singularity analysis of geochemically combined elements, and make a grid diagram. In the second step, using the local singularity raster of the appropriate selected elements, we use Formula (3) to carry out the overlay analysis:

$$[O] = \sum_i^n [\alpha_i], \tag{3}$$

where $i$ denotes proportionality element (such as Au, Ag, and so on), $n$ denotes types of geochemical combination elements, $[\alpha_i]$ denotes the local singular raster datasets of the $i$ element, and $[O]$ denotes geochemical composition element local singular value space overlay raster dataset, whose significance represents the anomaly map of the deposit type.

According to the geological basis and geochemical prospecting mark, a geochemical combined element model is established. According to the model of the geochemical combined element of the porphyry copper mine, the geological significance of the local singularity map is given.

### 3.3. Concentration–Area Model (C–A)

The C–A model, originally developed by [33], represents the first important step in fractal/multifractal modeling of geochemical data and has been "a fundamental technique for modeling of geochemical anomalies" [34]. The C–A fractal model gives:

$$A(\rho \leq v) \propto \rho^{-a_1}; \ A(\rho > v) \propto \rho^{-a_2}, \tag{4}$$

where $A(\rho)$ represents the area with concentrations greater than or equal to the contour value $\rho$, $v$ is the threshold, and $a_1$ and $a_2$ are fractal dimensions, which are greater than zero. These two fractal parameters can be estimated from the slopes of the best-fitting straight lines in the log–log plot of $A(\rho)$ versus $\rho$. $\propto$ denotes proportionality (is the mathematical symbol for "proportional to"). The C–A model can be used for raw point geochemical data, contour maps, or raster maps of elements. Two approaches can be used to calculate the enclosed area. One is based on the contour map created by interpolation procedures, and the other is based on superimposing a grid with cells on the study area and calculating the area by means of a box counting method. Distinct patterns, each corresponding to a set of similarly shaped contours, can be separated by different straight segments fitted to the values of the contours and enclosed areas on the log–log plot. The slopes of these straight lines can be taken as an estimation of the exponents of the power-law relation in Equation (4). The optimum threshold for separating geochemical anomalies from the background is the concentration value common to both linear relationships on the log–log plot [22,32]. Based on the deposit type diagram, the optimal threshold is determined quantitatively by the C–A method.

## 4. Results and Discussion

### 4.1. The Question of ilr-RPCA-Back clr

The extraction of geochemically combined elements has attracted much attention in geochemical prospecting, with various methods available. Currently, the methods of combination element identification are mainly PCA, RPCA, FA, RFA, and so on. These methods require the use of logarithmic transformations or logarithmic ratio transformations (alr, *clr*, *ilr*, and so on) to eliminate or attenuate the effect of component data closure as much as possible. Here, the widely used *ilr-RPCA-back clr* method is discussed.

Question 1: How does the process of element exchange impact the compositional variations of geochemical elements?

The *ilr* transform method itself belongs to the asymmetric transformation, and the correspondence between the variables before and after the transformation is disrupted. This disruption is primarily due to changes in the order of elements in the *ilr* formula. Consequently, when applying this method to stream sediment data, different combinations of elements can be obtained through the same transformation. Based on the experiments with the six elements related to the original data of the porphyry copper deposit, the influence of the order change of different elements on *ilr* transformation is discussed. As shown in the figure, the Ag, Mo, and W elements in Figure 3a represent the first principal component and exhibit strong positive correlations (Table 2). However, in Figure 3b, these same elements show poor representation and weak correlation (Table 2). In Figure 3c, the Ag-Mo-W and Au-Cu elements, respectively, demonstrate negative loadings on the first principal component and positive loadings on the second principal component (Table 2). Figure 3d reveals that Ag and Mo elements maintain a strong positive correlation as they represent the first principal component. From these results, it can be concluded that further research is required to investigate the application of this method for determining geochemical elements.

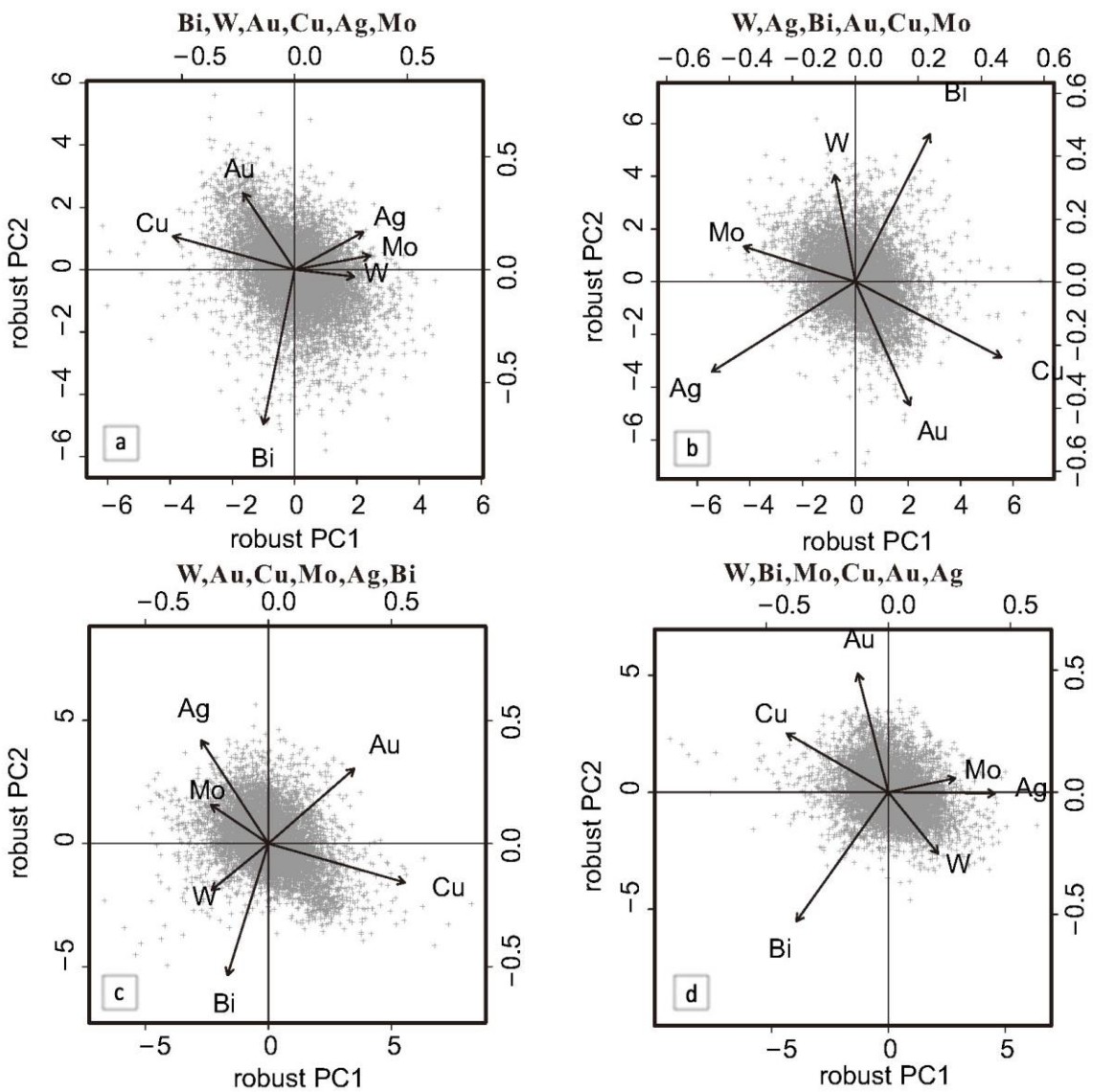

**Figure 3.** Biplots of different elements in sequence ((**a**): Bi, W, Au, Cu, Ag, Mo; (**b**): W, Ag, Bi, Au, Cu, Mo; (**c**): W, Au, Cu, Mo, Ag, Bi; (**d**): W, Bi, Mo, Cu, Au, Ag).

**Table 2.** ILR-RPCA-back CLR Statistical Table of different elements in sequence.

| Sequence 1 (Figure 3a) | | | Sequence 2 (Figure 3b) | | | Sequence 3 (Figure 3c) | | | Sequence 4 (Figure 3d) | | |
|---|---|---|---|---|---|---|---|---|---|---|---|
| Elements | PC1 | PC2 | Elements | PC1 | PC2 | Elements | PC1 | PC2 | Elements | PC1 | PC2 |
| Bi | 0.17 | 0.86 | W | −0.08 | 0.42 | W | 0.29 | 0.24 | W | 0.26 | −0.31 |
| W | −0.33 | 0.04 | Ag | −0.57 | −0.36 | Au | −0.44 | −0.38 | Bi | −0.47 | −0.66 |
| Au | 0.29 | −0.42 | Bi | 0.30 | 0.58 | Cu | −0.69 | 0.20 | Mo | 0.34 | 0.07 |
| Cu | 0.68 | −0.19 | Au | 0.22 | −0.49 | Mo | 0.29 | −0.20 | Cu | −0.52 | 0.30 |
| Ag | −0.38 | −0.21 | Cu | 0.58 | −0.30 | Ag | 0.34 | −0.52 | Au | −0.16 | 0.61 |
| Mo | −0.42 | −0.08 | Mo | −0.44 | 0.14 | Bi | 0.21 | 0.67 | Ag | 0.55 | −0.01 |
| Cumulative Proportion | 0.34 | 0.58 | Cumulative Proportion | 0.32 | 0.61 | Cumulative Proportion | 0.37 | 0.63 | Cumulative Proportion | 0.31 | 0.58 |

Question 2: Score and load transform to *clr* space; has the element correspondence changed?

$$clr(Y) = log \frac{x_i}{\sqrt[D]{\prod_{i=1}^{D} x_i}} (i = 1, 2, 3, \cdots, D-1, D), \tag{5}$$

$$u_i = \sqrt{\frac{i}{i+1}} \left[ \underbrace{\frac{1}{i}, \cdots, \frac{1}{i}}_{i \ \ elements}, -1, 0, \cdots, 0 \right] (i = 1, 2, 3, \cdots, D-1), \tag{6}$$

$$U = [u_1, u_2, \cdots, u_{D-2}, u_{D-1}]^T (i = 1, 2, 3, \cdots, D-1), \tag{7}$$

$$ilr(X) = \sqrt{\frac{i}{i+1}} ln \left[ \frac{\sqrt[i]{\prod_{j=1}^{i} x_j}}{x_{i+1}} \right] (i = 1, 2, 3, \cdots, D-1), \tag{8}$$

$$clr(Y) = ilr(X)U \tag{9}$$

Based on the information provided, this paper analyzes the reasons for the aforementioned problems, starting with the fundamental formula. Although the score and load can be projected into the clr space after the transformation between *clr* and *ilr*, the calculation steps of *ilr-RPCA-back clr* reveal the following issues:

Step 1: The original data matrix $A$ ($R \times D$), after the ILR transformation, finds that the last element has not been given the value of the ILR transform, so that the matrix $B$ ($R \times (D-1)$) of an element is reduced.

Step 2: The RPCA transformation of data matrix $B$ ($R \times (D-1)$) after *ilr* can only obtain $D-1$ principal components; at this time, the elements represented are only the first $D-1$ elements, the score is matrix $C$ ($R \times (D-1)$), and the load is matrix $D$ (($D-1) \times (D-1)$).

Step 3: $U$ transformation of matrix $C$ and $D$ transform to CLR space, $U$ score matrix ($R \times (D-1)$)·(($D-1) \times D$) to obtain a new score matrix $E$ ($R·D$), namely $U^T$·matrix ($D \times (D-1)$)·(($D-1) \times (D-1)$) obtain the new matrix $F$ ($D \times (D-1)$) and set up a corresponding contact element.

It is evident that the final element does not contribute to the principal component analysis, and consequently, the transformed load obtained using the transformation formula does not contain any information pertaining to this specific element. Furthermore, the relationships among elements in the transformed load are not identical to their original relationships but rather represent comprehensive associations. Additionally, it should be noted that the relationship between *clr* and *ilr* is $U$ instead of $U^T$. Hence, it is necessary to handle the correspondence between elements in the new load matrix appropriately. Since *ilr* is not a matrix, *clr* = *ilr*·*U* cannot be converted to "*clr* = $U^T$·*ilr*". So, for the final *clr* space inverse transformation, the only problem is the corresponding elements of the relationship.

This paper argues that the CLR, ILR, and RPCA do not have any problems, but the combination of the three should be considered carefully for the identification of geochemically combined elements and types of deposits. For this issue, this paper does not propose a definitive solution but would like to take this opportunity to raise the question and hope that the mathematical geologists will study deeply and solve this problem.

### 4.2. Selection of Element Association Associated with Porphyry Copper Mineralization

Since stream sediment data belong to compositional data, it is not appropriate to conduct relevant combinatorial analysis of the data without eliminating the closure effect.

Based on the original geochemical data, this paper is under the guidance of the study of geological laws, the establishment of geochemical markers, and the exploration of geochemical models (supergene). According to the porphyry deposits in the Gangdese metallogenic belt, part of the geochemical elements combined statistically have been found to be elements of the porphyry copper deposits in combination with Cu, Mo, Au, Ag, W, and Bi, which is convenient to distinguish from other types of ore deposits (Table 3).

**Table 3.** Composite element statistics table.

| Deposits and Metallogenic Belts | Geochemical Anomaly Element Combination | Sampling Mode | References |
|---|---|---|---|
| Xiong Cun | Cu, Au, Mo | regional geochemical anomalies | [35] |
| Xiong Cun | Cu, Au, Ag, Pb, Zn | soil anomaly | [36] |
| Ji Ru | Cu, Mo, W, Bi | regional geochemical anomalies | [35] |
| Zhu Nuo | Au, Cu, Mo, W | regional geochemical anomalies | [35] |
| Zhu Nuo | Cu, Mo, W, Au, Pb, Zn, Ag | stream sediment | [37] |
| Chong Jiang | Cu, Mo, Au, Ag, Pb, Zn, Hg, Sb | stream sediment | [37] |
| Chong Jiang | Cu, Mo, W, Bi, Pb, Ag | regional geochemical anomalies | [37] |
| Qu Long | Cu, Mo, W, Bi, Pb, Ag | stream sediment | [38] |
| Qu Long | Cu, Mo, W, Bi, Sn | regional geochemical anomalies | [5] |
| Jia Ma | Cu, Bi, Au, Ag, Pb, Zn | stream sediment | [39] |
| Jia Ma | Cu, Mo, Au, Ag, Bi, Sn | soil geochemistry | [39] |
| Gangdese polymetallic metallogenic belt | Cu, Mo, W, Au, Ag, Bi | geochemical anomaly | [40] |
| Gangdese polymetallic metallogenic belt | Cu-Mo, Au-Ag, Cu-Mo-Au, Cu-Au-Ag | combination geochemical anomaly | [40] |
| Gangdese porphyry copper deposit | Cu, Mo, Pb, Zn, Ag | | [41] |
| Gangdese copper polymetallic metallogenic belt | Cu, Au, Ag, W, Mo, Bi | geochemical anomaly | [42] |
| Gangdese copper polymetallic metallogenic belt | Cu-Mo, Cu, Cu-Mo-Au, Cu-Au | geochemical anomaly | [42] |
| statistical results | Cu(21), Mo(16), Au(14), Ag(12), W(8), Bi(8), Pb(7), Zn(5), Hg(1), Sb(1), Sn(1) | final choice | Cu(21), Mo(16), Au(14), Ag(12), W(8), Bi(8) |

### 4.3. Spatial Overlay Analysis of Geochemical Singularity Index α-Value of Porphyry Copper Deposit

Recent advances in identifying weak geochemical anomalies refer to the singularity mapping technique proposed by Cheng [16], and it has been demonstrated as a powerful multifractal tool for identifying weak geochemical anomalies in complex geological settings or areas covered by overburden [20,22,23,43]. The window-based method was used to calculate the local singularity index based on GeoDAS. Considering that the formation of porphyry copper deposits in the western Gangdese metallogenic belt is closely related to the specific geochemical composition elements, the local singularity analysis of individual elements cannot be used to meet the objective of recognizing the weak anomalies of porphyry copper deposits. Therefore, according to the geochemical composition elements of porphyry copper deposits, singularity analysis of each element has been carried out, and the local singular maps of porphyry copper deposits have been obtained by spatial overlay analysis. The spatial distribution map of α-A (Figure 4b) highlights the weak anomalies relative to α-Cu raster contour maps (Figure 4a). It is also shown that the spatial overlay singular values (α-A) are apparently enriched in the acid intrusive rocks and volcanic rocks and spread in the NW-trending direction according to the regional faults. The anomaly threshold values of α-A and α-Cu were obtained by the C–A model. Furthermore, the log–log plots of the concentration (α) versus the number of samples with concentration values greater than or equal to α are constructed to examine whether or not the distribution of A and Cu follows a fractal. It can be observed that two straight lines can be fitted (Figure 5), suggesting they may satisfy a multifractal distribution.

In order to facilitate the analysis of the advantages and disadvantages of the two methods, this paper presents a typical deposit analysis map (Figure 6) based on the known deposit and the abnormal range of the degree of fit. The analysis diagram clearly shows that the recognition degree of porphyry copper is obviously improved compared with the single element method (Figure 6(a1–a3, b1–b3)). For skarn-type copper deposits, there is a large area anomaly in the Ri a deposit. The reason is that the copper ore bodies in the Ri a deposit are produced in the porphyry bodies. Therefore, this method makes it difficult to identify or distinguish such deposits and porphyry copper mines. However, for skarn-type deposits unrelated to porphyry, the method demonstrates a certain degree of distinction, especially the Shesuo deposit (Figure 6(b5)). This method distinguishes porphyry molybdenum ore from the single element method, but both of them have anomalies, indicating that

these two methods have limited ability to distinguish porphyry copper and porphyry molybdenum. Although this method is not as effective as the single element method in distinguishing porphyry-type molybdenum ore, it still shows some abnormalities. In the case of porphyry gold, this method has the advantage of reducing the abnormal level compared with the single element method. For such deposits as Sa ka (Figure 6(a5,b5)) and Ri na (Figure 6(a6,b6)), the use of the single element method is divided into three abnormal zones, but the use of this method only allows for two levels of zoning. This advantage can be reduced in the early stages of abnormal evaluation of porphyry copper, reducing the input of such abnormal identification.

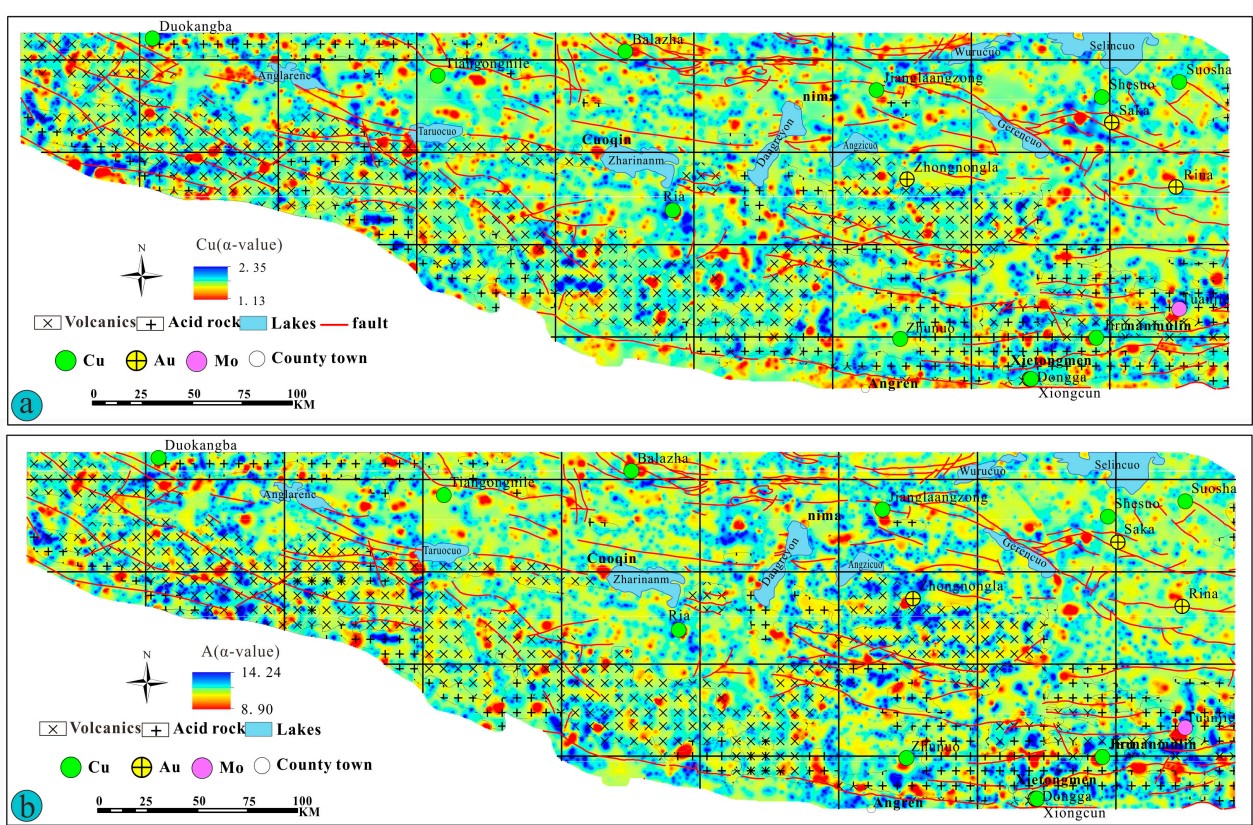

**Figure 4.** (**a**) Anomaly map of singular values of copper elements; (**b**) singular value anomaly map of porphyry copper deposit geochemical composition element.

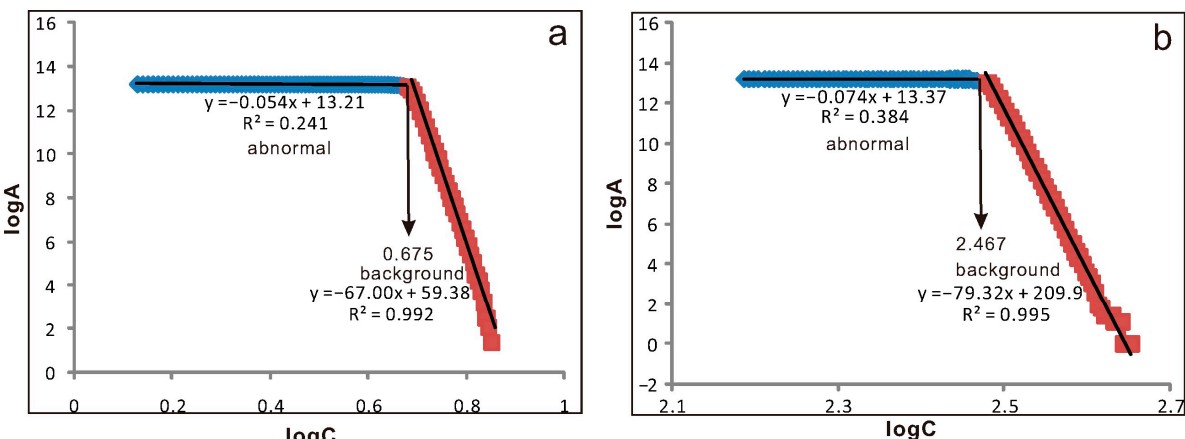

**Figure 5.** (**a**) Log–log plot of C–A fractal model of copper element; (**b**) log–log plot of C–A fractal model of porphyry copper mineral combination elements.

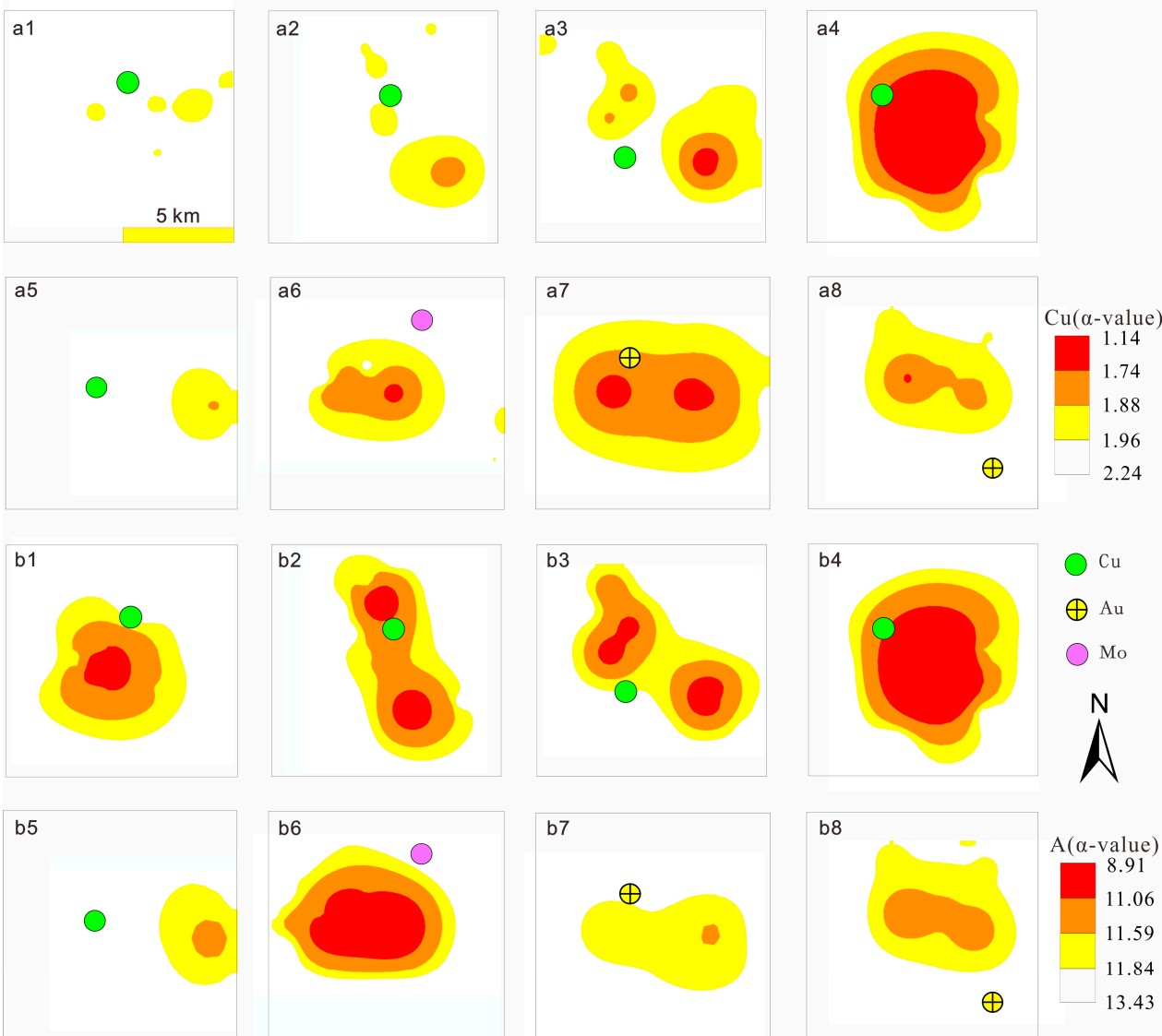

**Figure 6.** Geochemical anomaly map of typical deposits in western Gangdese. (**a1**–**a8**) Singular anomalies of copper elements of Zhu nuo, Jianglaang-zong, Balazha, Ri a, She suo, Tuan jie, Sa ka, and Ri na; (**b1**–**b8**) Singular anomalies of association elements of Zhu nuo, Jianglaangzong, Balazha, Ri a, She suo, Tuan jie, Sa ka, and Ri na; porphyry Cu deposit: Zhu nuo, Jianglaangzong, and Balazha; skarn Cu deposit: Ri a and She suo; porphyry Mo deposit: Tuan jie; porphyry Au deposit: Sa ka and Ri na.

## 5. Conclusions

The results of this study led to the following conclusions.

1.  The *ilr*-RPCA-back CLR model has two issues. (1) The change in element position severely affects the relationship between geochemical elements. (2) The score and load transformation to clr space disrupts the corresponding relationship between the elements. Therefore, it is important to carefully consider the use of this model for the identification of geochemical element combinations.
2.  The proposed method of singular value overlay analysis has clear advantages in identifying porphyry copper deposits. However, it is difficult to distinguish skarn-type copper related to porphyry from porphyry molybdenum. Additionally, the distinction between porphyry skarn-type copper deposits and porphyry gold deposits is not well defined. Nevertheless, this method can reduce the anomaly grade.

3. This paper investigates the geochemical laws, geochemical markers, and geochemical models based on geological foundations. It provides objective geological connotations for the identification and evaluation of anomalies using geochemical data. By overcoming the limitations of traditional technical methods and single element analysis, which are influenced by elemental chemical properties, redox environment, weathering erosion, and other factors, these methods offer significant advantages in anomaly screening. They greatly reduce multiple solutions and subjectivity, highlighting the prospecting value of anomalous regularity.

**Author Contributions:** Writing—original draft, S.Z.; writing—review and editing, X.J. and S.G.; visualization, S.Z.; investigation, S.G.; methodology, X.J. and S.Z. All authors have read and agreed to the published version of the manuscript.

**Funding:** This research was funded by the National Natural Science Foundation of China (No. U22A20572).

**Conflicts of Interest:** The authors declare no conflict of interest.

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
