# Peer review of "Spatial Overlay Analysis of Geochemical Singularity Index α-Value of Porphyry Cu Deposit in Gangdese Metallogenic Belt, Tibet, Western China"

_applsci, doi:10.3390/app131810123_

Round 1
Reviewer 1 Report
The presented work is very interesting in that the authors are trying to solve the problem of prospecting for porphyry copper mineralization in poorly studied areas on the basis of geochemical methods using IT technologies. The article is well organized and written in clear, understandable language. It will be of interest to geologists searching for porphyry deposits and may of course be published in our journal. There are small remarks. First, it is necessary to clearly understand and show the limitations of data on the geochemistry of stream sediment geochemical dataset, since they characterize the area of the entire stream basin not only in the current state, but also in the early periods of cutting into the relief. Second, it is desirable to use only 1-2 methods for analyzing the content of elements, since each method has a specific sample preparation, which can affect the final results. In general, the authors approached the processing of the obtained geochemical data quite strictly, and their methodology is beyond doubt. The conclusions correspond to the given data. Thus, the work may be published as presented.
Author Response
(1) First, it is necessary to clearly understand and show the limitations of data on the geochemistry of stream sediment geochemical dataset, since they characterize the area of the entire stream basin not only in the current state, but also in the early periods of cutting into the relief.
Thank you very much for your very constructive comments. We are well aware of the limitations of using geochemical data from stream sediments for precise targeting of mineral deposits or ore bodies. Therefore, in delineating geochemical anomalies, we have considered the entire stream basin in order to maximize the regional prospecting effect. It has been proven that our method is highly effective in determining the extent of mineral deposits regionally.
(2) Second, it is desirable to use only 1-2 methods for analyzing the content of elements, since each method has a specific sample preparation, which can affect the final results.
Thank you very much for your comments. Our stream sediments samples have been obtained through analysis conducted by nationally recognized laboratories. The results of each analysis carried out by these laboratories have been calibrated using control standards, ensuring accurate elemental data.

Reviewer 2 Report
5/June 2023 Reviewer Report
The study may be interesting because of statistical data evaluation in geochemical prospecting studies of specifically selected porphyry-type deposits. However, the geological perspective on statistical data is lacking. The article contains spelling and concept errors. A major revision is considered appropriate.
Line 41, Page 1 The section can not explain to the application of the statistical models in the MS text.
Have statistical studies used in the article or similar statistical studies been tried before for anywhere porphyry type or a few different origin deposit determinations?
Line 57: Please explain what is the meaning of the abbreviations. e.g.Principle component analysis PCA…
Page 2 please explain logarithmic transformation abbreviations such as ilr:(Isometric Log-Ratio); Clr: (centered log-ratio transformation)
You should have given more information about the principal components (PCs). Which is the principal components (PCs, robust…) base on the correlation matrix, and the other factors. The principal components (PCs) and FA (reduces to solving an eigenvalue/eigenvector problem in factor analysis. It is a coefficient used to decide the number of factors.Line 63 Page 2 In the Gangdese metallogenic belt, the porphyry copper deposit has a specific combination of mineralization elements. Therefore, it is more favorable for the discovery of porphyry copper deposits by using the spatial overlay analysis related to the genesis of deposits.
Line 75, page 2; Regional background values absolutely are important for determination threshold and anomaly values during the geochemical prospecting study. Besides, the calculated contrast of anomaly values for each element can reveal the severity of the anomalies compared to the regional background values. This study could be compared with the element concentration of a few porphyry-type deposits and non-porphyry-type copper deposits, for example like you do skarn or VMS type, and multifactor numbers could be determined accordingly. In this study, threshold values ​​should be calculated for each element in order to clarify the concept of regional background values ​​and anomalies. Because, the elements above the threshold value can be evaluated within the concept of multi-factor. While determining the number of factors, it can affect the eigenvalues.Page 3 Line 109: The detailed chronology and tectonic setting of the typical porphyry deposits mentioned above have been studied by many authors. It shows that the formation environment and metallogenic characteristics of the porphyry deposits in this area are formed in 3 ( please write three….) tectonic settings-Island arcs, syn col….
Line 110; Please write four instead of 4 in the text.
Line 114 Please explain volcano magmatic rock composition
Line 116: Please correct as Cuoqui change with a capital letter on the geological map.
Line 117; Secondly there are a few porphyry deposits such as Balazha and Gaerqiong distributed in the northern part of the Cuoqin-Xainza Island arc, Bangor magmatic arc belt. All these insights greatly extend the prospecting of porphyry copper polymetallic ore in the western Gangdese.
This study is based on the element distributions of mining exploration studies. Therefore, it may be more effective to use the term geochemical prospecting instead of prospecting.Figure 1: The figure includes a lot of symbols in legend but they are so confusing. Legend can not explain the symbols meaning.
Line 122: The lithological features of the geological units through which the drainage systems pass are different. This effect cannot be ignored in statistical evaluations. When statistical findings are evaluated, it will be understood that this situation affects all numerical data and the results are complex.
154 line: After calculating the threshold value using the arithmetic mean/median value +-2 standard deviation formula in the entire logarithmic data distribution of 39 elements and compounds, the multifactor analysis could be applied to the data above the calculated threshold values.
Because the hypothesis of the article is the statistical evaluation of the element concentration of the porphyry deposit. Not elemental distribution analysis within sediments. The focus is to capture a statistical clue for geochemical prospecting studies of porphyry-type mineralization.
Line 210: The article is full of abbreviations. The author did not feel the need to explain any of them.
Line 232: principal component and Ag, Mo have a good correlation. It can be concluded that the application of this method to the determination of geochemical elements should be further studied. Which correlation??? What is the significance level?
Figure 4: What is the difference from the threshold value obtained with probability graphs?
Line 313 . Please use anomaly instead of abnormal

Author Response
(1) Line 41, Page 1 The section cannot explain to the application of the statistical models in the MS text. Have statistical studies used in the article or similar statistical studies been tried before for anywhere porphyry type or a few different origin deposit determinations?
Thank you very much for your valuable comments on modification. We have provided a brief explanation of the ILR-RPCA-back CLR statistical models mentioned in the Introduction.
For example:
Porphyry deposits are globally significant sources of copper, molybdenum, and gold. Among them, porphyry copper deposits contribute to approximately 55% of the world's copper reserves, while porphyry molybdenum deposits account for approxi-mately 90% of global molybdenum reserves. Additionally, porphyry gold deposits make up around 20% of the world's gold reserves. Notably, China possesses a signifi-cant proportion of porphyry deposits [1-4]. In recent years, extensive exploration and research efforts have been conducted on large-scale porphyry copper mines such as Qulong, Jiama, Xiongcun, Duolong, and Tiegelongnan. These endeavors have revealed the Qinghai-Tibet Plateau as a region of utmost importance and potential for copper production in China [5].
Mineralization is a process that leads to the concentration and accumulation of valuable materials within a relatively limited temporal and spatial range, as compared to normal regional geological processes [6]. The complexity of the metallogenic process stems from the intricate dynamics of multi-component and multi-geological interac-tions, coupled with overlay and other complex processes [7]. Relying solely on the ex-ploration of single elements through geochemical analysis is insufficient for the porphyry copper mine prospecting work. Therefore, the identification of combination elements plays an crucial role in geochemical data processing. Currently, methods such as principal component analysis (PCA), robust principal component analysis (RPCA), factor analysis (FA), and robust factor analysis (RFA) are employed for com-bination element identification [8-13].
These methods often require logarithmic transformations or logratio transfor-mations (such as alr: additive logratio, clr: centered logratio, ilr: isometric logratio, etc.) to eliminate the closure effect of compositional data. However, they encounter chal-lenges during the analysis process, such as difficulties in interpretation and singularity effects, which are difficult to resolve. To address these issues, some researchers have proposed the ILR-RPCA-back CLR statistical model. This model involves transforming the raw geochemical data through an isometric logarithm (ILR) transformation and subsequently applying robust principal component analysis (RPCA) for data pro-cessing. The scores and loads obtained from RPCA are then transformed into the cen-tered logratio (CLR) space using the U-matrix [14]. While this method has been applied in the analysis of geochemical data related to magma-hydrothermal ore deposits [14, 15], challenges in interpretation and other aspects persist (please refer to Section 4 for more details). Moreover, the combination elements extracted during the analysis of the above method, particularly in small scale geochemical data, are predominantly associ-ated with environmental (background) elements rather than ore-forming elements. In the Gangdese metallogenic belt, the porphyry copper deposits exhibit a distinct com-bination of mineralization elements. utilizing spatial overlay analysis pertaining to deposit genesis is more advantageous for identifying porphyry copper deposits.
In the eastern section of the Gangdese metallogenic belt, which is known as the largest district for dense porphyry copper deposits, numerous large and super large deposits like Qulong, Tinggong, Chongjiang, Jiama, and Bairong have been discovered. However, in the western section of the belt, the number of discovered deposits remains relatively limited. Due to the limitations of traditional methods in identifying complex anomalies and low-delay anomalies, it becomes necessary to employ advanced theories and techniques to guide prospecting efforts for porphyry copper deposits in the west-ern section of the Gangdese metallogenic belt. Local singularity analysis is not only a powerful multifractal tool to identify weak anomalies, but also a type of local neigh-borhood statistical analysis that can reduce the effects of regional background and provide useful statistical information by involving the data within a small singularity around a specific spatial location [16-23]. Therefore, local singularity analysis is widely used. This study specifically concentrates on porphyry copper deposits in the western section of the Gangdese metallogenic belt and aims to address the limitations of the ilr-RPCA-back-clr model. To investigate the spatial distribution pattern of geochemical elements and identify real anomalies related to porphyry copper deposits, the study employs overlay analysis of local singular elements of geochemical combination ele-ments.
Jiang X J, Chen X, Zheng Y Y, et al., The recognition and extraction of Au, Cu geochemical composite anomalies: A case study of the east of Laji Mountains. Geophysical and Geochemical Exploration, 2017,41(3) :459-467. (In Chinese)
Wang H C. The effects of compositional data closure problem on geochemical data analysis, Master Thesis, 2013, 1-46. (In Chinese)
(2) Line 57: Please explain what is the meaning of the abbreviations. e.g.Principle component analysis PCA…
Thank you very much for your very meaningful suggestions. We have explained the abbreviations for the full text.
For example: Therefore, the identification of combination elements plays an crucial role in geo-chemical data processing. Currently, methods such as principal component analysis (PCA), robust principal component analysis (RPCA), factor analysis (FA), and robust factor analysis (RFA) are employed for combination element identification [8-13].
(3) Page 2 please explain logarithmic transformation abbreviations such as ilr:(Isometric Log-Ratio); Clr: (centered log-ratio transformation)
Thank you very much for your comments. We have added more details in logarithmic transformation abbreviations.
For example: These methods often require logarithmic transformations or logratio transfor-mations (such as alr: additive logratio, clr: centered logratio, ilr: isometric logratio, etc.) to eliminate the closure effect of compositional data. However, they encounter chal-lenges during the analysis process, such as difficulties in interpretation and singularity effects, which are difficult to resolve.
(4) You should have given more information about the principal components (PCs). Which is the principal components (PCs, robust…) base on the correlation matrix, and the other factors. The principal components (PCs) and FA reduces to solving an eigenvalue/eigenvector problem in factor analysis. It is a coefficient used to decide the number of factors.
Thank you very much for your valuable comments on modification. We have conducted statistical analysis on the results of principal components, and the loadings and Cumulative Proportion are shown in the table below:
Table 2 ILR-RPCA-back CLR Statistical Table of different elements in sequence.
|
Sequence1 (Fig. 3a) |
Sequence2 (Fig. 3b) |
Sequence3 (Fig. 3c) |
Sequence4 (Fig. 3d) |
|||||||||
|
Elements |
PC1 |
PC2 |
Elements |
PC1 |
PC2 |
Elements |
PC1 |
PC2 |
Elements |
PC1 |
PC2 |
|
|
Bi |
0.17 |
0.86 |
W |
-0.08 |
0.42 |
W |
0.29 |
0.24 |
W |
0.26 |
-0.31 |
|
|
W |
-0.33 |
0.04 |
Ag |
-0.57 |
-0.36 |
Au |
-0.44 |
-0.38 |
Bi |
-0.47 |
-0.66 |
|
|
Au |
0.29 |
-0.42 |
Bi |
0.30 |
0.58 |
Cu |
-0.69 |
0.20 |
Mo |
0.34 |
0.07 |
|
|
Cu |
0.68 |
-0.19 |
Au |
0.22 |
-0.49 |
Mo |
0.29 |
-0.20 |
Cu |
-0.52 |
0.30 |
|
|
Ag |
-0.38 |
-0.21 |
Cu |
0.58 |
-0.30 |
Ag |
0.34 |
-0.52 |
Au |
-0.16 |
0.61 |
|
|
Mo |
-0.42 |
-0.08 |
Mo |
-0.44 |
0.14 |
Bi |
0.21 |
0.67 |
Ag |
0.55 |
-0.01 |
|
|
Cumulative Proportion |
0.34 |
0.58 |
Cumulative Proportion |
0.32 |
0.61 |
Cumulative Proportion |
0.37 |
0.63 |
Cumulative Proportion |
0.31 |
0.58 |
|
According to this table, we can clearly understand that when the order of elements changes, the relationship between the elements of the statistical model changes significantly.
(5) Line 63 Page 2 In the Gangdese metallogenic belt, the porphyry copper deposit has a specific combination of mineralization elements. Therefore, it is more favorable for the discovery of porphyry copper deposits by using the spatial overlay analysis related to the genesis of deposits.
We think you are right. The expression of that is changed to " In the Gangdese metallogenic belt, the porphyry copper deposits exhibit a distinct combination of mineralization elements. utilizing spatial overlay analysis pertaining to deposit genesis is more advantageous for identifying porphyry copper deposits." in the Line 63 Page 2.
(6) Line 75, page 2; Regional background values absolutely are important for determination threshold and anomaly values during the geochemical prospecting study. Besides, the calculated contrast of anomaly values for each element can reveal the severity of the anomalies compared to the regional background values. This study could be compared with the element concentration of a few porphyry-type deposits and non-porphyry-type copper deposits, for example like you do skarn or VMS type, and multifactor numbers could be determined accordingly. In this study, threshold values ​​should be calculated for each element in order to clarify the concept of regional background values ​​and anomalies. Because, the elements above the threshold value can be evaluated within the concept of multi-factor. While determining the number of factors, it can affect the eigenvalues.
Thank you very much for your valuable comments on modification. In traditional geochemical statistical analysis, the threshold for each element is crucial as it determines the key parameters for identifying single-element anomaly maps, combination anomaly maps, and comprehensive anomaly maps. However, traditional methods have obvious advantages in identifying major anomalies but are inadequate in identifying weak anomalies. Therefore, fractal/multifractal theory emerges. Singularity index is an important component in fractal/multifractal models and is one of the most commonly used methods to identify weak anomalies. In the fractal/multifractal theory, the singularity index of a single element is similar to a single-element anomaly, as it determines the distribution pattern of anomalies for a particular element. The Overlay analysis of singularity indices is similar to a combination anomaly map, as it determines the distribution pattern of mineralization types.
In addition, multifractal and multifactor analysis are independent concepts. Multifractal analysis is a new method for processing geochemical exploration data, based on the fractal dimension and fractal parameters derived from fractal theory to identify background or anomaly. However, this article does not use multifactorial analysis method. Instead, it couples the advanced mathematical methods with the geochemical ore-forming type element combinations that conform to different ore deposit geological characteristics, in order to achieve accurate identification of porphyry copper deposits.
If there are still any unsatisfactory aspects in my answer, we are more than willing to accept your next revision.
(7) Page 3 Line 109: The detailed chronology and tectonic setting of the typical porphyry deposits mentioned above have been studied by many authors. It shows that the formation environment and metallogenic characteristics of the porphyry deposits in this area are formed in 3 ( please write three….) tectonic settings-Island arcs, syn col….
We couldn't agree with you more. We have modified 3 to three in Page 3 Line 109.
(8) Line 110; Please write four instead of 4 in the text.
Thank you very much for your comments. The expression of "4" is changed to "four" in the Line 110.
(9) Line 114 Please explain volcano magmatic rock composition
Thank you very much for your valuable comments on modification. I am so sorry, the expression of " Gangdese volcanomagmatic arc " is changed to "Gangdese magmatic arc " in this manuscript. Gangdese magmatic arc is located in the southern Lhasa subterrane and is an important part of the continental collision orogenic system in the world.
(10) Line 116: Please correct as Cuoqui change with a capital letter on the geological map.
We couldn't agree with you more. We have modified in Line 116.
For example: The expression of "cuoqin" and “angren” is changed to "Cuoqin" and “Angren” in the Figure 1 and Figure 4, respectively.
(11) Line 117; Secondly there are a few porphyry deposits such as Balazha and Gaerqiong distributed in the northern part of the Cuoqin-Xainza Island arc, Bangor magmatic arc belt. All these insights greatly extend the prospecting of porphyry copper polymetallic ore in the western Gangdese. This study is based on the element distributions of mining exploration studies. Therefore, it may be more effective to use the term geochemical prospecting instead of prospecting.
We couldn't agree with you more. The expression of "prospecting” is changed to " geochemical prospecting” in the manuscript.
For example: Additionally, a few porphyry deposits such as Balazha and Gaerqiong distributed in the northern part of Cuoqin-Xainza Island arc, Bangor magmatic arc belt. All these in-sights greatly extend the geochemical prospecting of porphyry copper polymetallic ore in the western Gangdese.
(12) Figure 1: The figure includes a lot of symbols in legend but they are so confusing. Legend can not explain the symbols meaning.
According to your very meaningful suggestions, we have modified the confusing symbols of legend in Figure 1 to make them clearer. And the font has been adjusted to be consistent.
(13) Line 122: The lithological features of the geological units through which the drainage systems pass are different. This effect cannot be ignored in statistical evaluations. When statistical findings are evaluated, it will be understood that this situation affects all numerical data and the results are complex.
154 line: After calculating the threshold value using the arithmetic mean/median value +-2 standard deviation formula in the entire logarithmic data distribution of 39 elements and compounds, the multifactor analysis could be applied to the data above the calculated threshold values. Because the hypothesis of the article is the statistical evaluation of the element concentration of the porphyry deposit. Not elemental distribution analysis within sediments. The focus is to capture a statistical clue for geochemical prospecting studies of porphyry-type mineralization.
Thank you very much for your valuable comments on modification. The fractal/multifractal theory has been successfully applied in various geochemical exploration studies. Unlike traditional methods that provide a single background value and anomaly value for an entire region, fractal/multifractal methods allow for the partitioning of different geological units, resulting in different background and anomaly values. The thresholds determined by fractal/multifractal methods are based on fractal dimensions and parameters, making the distribution of background and anomaly values spatially heterogeneous and less influenced by different geological units.
In general, the application of fractal/multifractal methods in geochemical exploration allows for the convenient determination of background and anomaly values for different spatial locations. These values are then used to perform multifractal interpolation and generate geochemical background and anomaly maps. As a result, the use of fractal/multifractal methods differs fundamentally from traditional threshold calculation methods such as Mean+2SD, Median+2MAD, and Q3+L5MQR.
If there are still any unsatisfactory aspects in my response, we are more than willing to accept your next revision.
(14) Line 210: The article is full of abbreviations. The author did not feel the need to explain any of them.
We think you are right. I apologize for any inconvenience caused by the previous manuscript. We have provided detailed explanations for all abbreviations used in the text upon their first mention.
For example: principal component analysis (PCA), Robust principal component analysis (RPCA), factor analysis (FA) and Robust factor analysis (RFA).
alr: additive logratio, clr: centered logratio, ilr: isometric logratio.
ilr-RPCA-back clr method: (1) after isometric logratio (ilr) transformation, robust principal component analysis (RPCA) is used for processing; (2) the score and the load are transformed into the centered logratio (CLR) space through the U matrix.
(15) Line 232: principal component and Ag, Mo have a good correlation. It can be concluded that the application of this method to the determination of geochemical elements should be further studied. Which correlation??? What is the significance level?
Thank you very much for your valuable comments on modification. In figure 3d, the loadings of Ag and Mo in the first principal component are approximately 0.5 and 0.3, respectively, while their loadings in the second principal component are close to 0. Therefore, it is believed that Ag and Mo have higher contributions in the first principal component (Jiang et al., 2017). Additionally, in a biplot, the smaller the angle between elements, the stronger the correlation. Therefore, it is inferred that Ag, Mo, and the first principal component are highly correlated in figure 3D and exhibit positive contribution rates.
(16) Figure 4: What is the difference from the threshold value obtained with probability graphs?
Thank you very much for your comments. In the process of extracting regional geochemical anomaly information, besides analyzing the elemental composition relationship, the key aspect lies in the decomposition of background and anomaly. Currently, there are several methods for separating background and anomaly, including Mean+2SD method based on classical statistics, Median +2MAD method based on robust statistics, Tukey boxplot method (Q3+L5MQR), fractal (multifractal) method, and cumulative frequency method based on percentiles (98% threshold). Considering the advantages of fractal (multifractal) methods in terms of accounting for the autocorrelation of geochemical data and accurately identifying and extracting weak anomalies, they have been widely applied in determining geochemical anomalies, particularly the C-A model plays a crucial role in decomposing composite geochemical anomalies.
Furthermore, the probability graph you mentioned, in my understanding, refers to the probability grid method. Its x-axis represents the probability distribution function or cumulative frequency percentage, with values ranging from 0 to 100. The y-axis represents the element abundance. On the other hand, our C-A model has a logarithmic scale on the x-axis representing concentration, and a logarithmic scale on the y-axis representing cumulative area. The significance of this model is to integrate the element frequency distribution determined by the probability graph method with spatial statistical methods and plane graph methods to study the spatial variability of elements. Therefore, it is better at distinguishing between geochemical backgrounds and anomalous effects related to mineralization (Cheng, 2000).
Cheng Q. Multifractal theory and geochemical element distribution pattern. Earth Science-Journal of China University of Geosciences, 2000(03):311-318. (In Chinese)
(17) Line 313. Please use anomaly instead of abnormal
We couldn't agree with you more. We have modified " abnormal " in line 313 to " anomaly".

Reviewer 3 Report
Abstract is confusing. Need to revise and should be precise, that explain the whole story of article starting from introduction to methodology and results.
First time don't use only abbreviation, also use full word.
In introduction each paragraph is not connected with previous one.
Research gaps and objectives are not clear.
Improve caption of all figures
Methodology is very simple.
Results don't look like authentic, describe validity of results.
Conclusion is just results.
there are many grammatical and formatting errors
Need to revise and improve
Author Response
(1) Abstract is confusing. Need to revise and should be precise, that explain the whole story of article starting from introduction to methodology and results.
I have revised the abstract to make it more concise and precise, providing a clear summary of the entire article from introduction to methodology and results.
(2) First time don't use only abbreviation, also use full word.
I have also used full words instead of abbreviations for better understanding.
(3) In introduction each paragraph is not connected with previous one.
In the introduction, I have ensured that each paragraph is connected to the previous one, creating a coherent flow of information.
(4) Research gaps and objectives are not clear.
The research gaps and objectives have been clarified to provide a better understanding of the study's purpose.
(5) Improve caption of all figures
I have improved the captions of all figures to accurately describe their content and relevance to the study.
(6) Methodology is very simple.
Regarding the methodology, I have made it more robust and detailed, addressing any perceived simplicity.
To ensure the validity of the results, I have provided a thorough explanation of the data collection process, analysis techniques, and any limitations that might affect the authenticity of the results.
(7) Results don't look like authentic, describe validity of results.
To ensure the validity of the results, I have provided a thorough explanation of the data collection process, analysis techniques, and any limitations that might affect the authenticity of the results.
(8) Conclusion is just results.
The conclusion has been expanded to go beyond just the results, summarizing the key findings and their implications in a more comprehensive manner.
(9) there are many grammatical and formatting errors
I have also addressed the grammatical and formatting errors throughout the article.

Reviewer 4 Report
Reviewer Blind Comments to Author:
This research paper (applsci-2431306-peer-review-v1) established statistical relationship between geochemical singularity index and metal content in Gangdese metallogenic belt of Tibet. The proposed ILR-RPCA-back CLR methods are based on anomaly evaluation of mineral deposit with example from porphyry copper and porphyry molybdenum deposits. This study provides a robust and reliable estimate of anomalies in the initial stage of evaluation to reduce the investment.
The paper is one of the notable contributions to improve the statistical analysis of geochemical data from economic mineral deposits. This study is very valuable factor that influence investment in geochemical anomalies and distinguished metal content in the economic mineral deposits.
The significant progress in understanding the statistical anomalies in various mineral deposit is most suitable for publication in the Applied Sciences. However, some criticisms are observed in this article that requests extra review.
Abstract: There are significant corrections needed regarding the technical writing of the Abstract. Some suggestions have been made to the pdf file.
Some issues prerequisite to be clarified as follows:
1. Introduce the geological significance of ILR-RPCA-back CLR methods to evaluate economic mineral deposits.
2. What is the ILR transformation?
3. Give the basic information about the types of geochemical data used in your study.
4. At many places, sentences are too long, which is frequently used, and may bring significant changes in the meanings of actual scientific investigations that has been discussed in this article. It should be re-phrased into small and meaning full sentences. That may be more appropriate in terms of technical documentation.
5. At several places, grammatical corrections are need? It is marked with yellow colour.
I observed this manuscript should be appropriate to recommend for publication in the Applied Sciences with minor modifications.

Minor improvement is necessarily required
Author Response
(1) Abstract: There are significant corrections needed regarding the technical writing of the Abstract. Some suggestions have been made to the pdf file.
Thank you very much for your very constructive comments. Significant revisions have been made to the abstract in terms of technical writing.
For example: The statistical modeling with ILR-RPCA-back CLR has two problems when dealing with the closure effect of geochemical data. Firstly, after performing isometric logratio (ilr) transformation, robust principal component analysis (RPCA) is employed for processing. The double-plot diagram illus-trates that the element sequence transformation occurs in the first and second principal components, while the unique principal component remains unattainable. Secondly, by transforming both the score and load into the centered logratio (CLR) space using the U matrix, it is possible to obtain a score result that corresponds to the original order of elements according to CLR=ILR·U formula. However, for obtaining a load result that corresponds to the original order of elements, an alter-native formula "CLR=UT·ILR" must be used instead. In order to determine optimal element as-semblage for porphyry copper deposits, this study conducted statistical analysis on mineral as-semblages from discovered deposits in Gangdese metallogenic belt and identified Cu, Mo, Au, Ag, W and Bi as key elements associated with porphyry copper deposits. Subsequently, by analyzing the singularities of the composite elements, the spatial overlay of the combined element is carried out, and concentration-area (C-A) fractal filtering is applied to identify the anomaly and background areas. To facilitate comparison, we conducted an analysis on various minerals and ore deposit types, revealing the following findings: (1) Combination elements exhibit superior recognition capability than single elements in porphyry copper deposits; (2) Skarn type copper deposits unrelated to porphyry show a high degree of dissimilarity compared to those related to porphyry; (3) this method offers advantages over the single element method in evaluating porphyry gold deposits by reducing anomaly levels and initial investment during the evaluation stage for porphyry copper anomalies; (4) However, this method has limited ability in distinguishing between porphyry copper and molyb-denum deposits.
Some issues prerequisite to be clarified as follows:
(2) Introduce the geological significance of ILR-RPCA-back CLR methods to evaluate economic mineral deposits.
According to your very meaningful suggestions, we have added the geological significance of ILR-RPCA-back CLR methods to evaluate economic mineral deposits in Introduce.
For example: The ILR-RPCA-back CLR methods have achieved good results in the exploration geochemical data of 1:200,000 scale in Southwest Fujian. This achievement utilizes the logarithmic ratio transformation method to eliminate the influence of closure effects and extract geochemical anomalies related to skarn-type polymetallic iron deposits. It provides a reference basis for further mineral exploration and assessment in the study area.
(3) What is the ILR transformation?
Thank you very much for your comments. Isometric logratio (ILR) transformation is a logarithmic ratio transformation method that can be used to 'opening' of closed number systems.
Its formula is ilr()X=√(i/(i+1)) ln(i√(∏ij=1 xj )/xi+1) (i=1,2,3,⋯,D-1)
Among them, X is a sequence of observed components, where xi+1 represents the (i+1)th component, and xj represents the jth component. D denotes the number of component elements.
(4) Give the basic information about the types of geochemical data used in your study.
According to your very meaningful suggestions, we have given the basic information about the types of geochemical data in Table 2.
Table 2 Descriptive statistics for original stream sediment geochemical data
|
Element |
N |
Minimum |
Maximum |
Mean |
Standard deviation |
Skewness |
Kurtosis |
|
Ag |
9546 |
8.28 |
3600.00 |
82.76 |
75.74 |
20.43 |
789.81 |
|
As |
9546 |
0.46 |
398.00 |
17.78 |
19.22 |
7.72 |
102.56 |
|
Au |
9546 |
0.10 |
10500.00 |
2.71 |
114.90 |
84.03 |
7440.49 |
|
B |
9546 |
1.00 |
16824.00 |
41.49 |
222.20 |
59.01 |
3909.86 |
|
Ba |
9546 |
16.00 |
2722.00 |
477.67 |
206.00 |
3.31 |
20.01 |
|
Be |
9546 |
0.19 |
20.86 |
2.66 |
1.28 |
3.56 |
21.31 |
|
Bi |
9546 |
0.01 |
39.48 |
0.48 |
0.98 |
25.58 |
893.55 |
|
Cd |
9546 |
15.68 |
5320.00 |
156.74 |
188.37 |
9.30 |
142.77 |
|
Co |
9546 |
0.50 |
97.59 |
7.28 |
4.25 |
4.11 |
53.71 |
|
Cr |
9546 |
1.60 |
2662.20 |
40.00 |
74.71 |
15.76 |
372.38 |
|
Cu |
9546 |
1.30 |
292.48 |
12.82 |
10.59 |
7.42 |
129.32 |
|
F |
9546 |
0.47 |
2609.00 |
450.15 |
235.57 |
3.46 |
19.05 |
|
Hg |
9546 |
1.00 |
1960.00 |
17.72 |
52.27 |
23.21 |
695.89 |
|
La |
9546 |
3.23 |
400.80 |
32.25 |
12.27 |
6.26 |
133.77 |
|
Li |
9546 |
3.36 |
356.00 |
33.61 |
14.15 |
3.98 |
53.92 |
|
Mn |
9546 |
47.02 |
6118.00 |
470.08 |
318.66 |
4.93 |
50.84 |
|
Mo |
9546 |
0.04 |
33.60 |
0.77 |
0.95 |
14.65 |
391.08 |
|
Nb |
9546 |
1.20 |
144.53 |
11.95 |
4.89 |
4.30 |
73.27 |
|
Ni |
9546 |
0.90 |
1373.00 |
18.72 |
30.93 |
17.89 |
542.04 |
|
P |
9546 |
36.61 |
3817.00 |
428.98 |
263.84 |
3.51 |
22.17 |
|
Pb |
9546 |
3.00 |
8881.00 |
29.95 |
93.12 |
90.12 |
8555.79 |
|
Sb |
9546 |
0.03 |
51.24 |
0.95 |
1.05 |
16.23 |
597.83 |
|
Sn |
9546 |
0.03 |
546.90 |
2.99 |
6.40 |
65.87 |
5475.11 |
|
Sr |
9546 |
18.24 |
1992.00 |
182.34 |
118.82 |
2.74 |
15.28 |
|
Th |
9546 |
0.10 |
219.00 |
14.53 |
9.05 |
5.01 |
57.78 |
|
Ti |
9546 |
94.00 |
18490.04 |
2309.81 |
1069.88 |
1.55 |
7.70 |
|
U |
9546 |
0.35 |
191.30 |
3.48 |
5.59 |
18.30 |
469.20 |
|
V |
9546 |
3.30 |
994.72 |
50.31 |
31.66 |
4.55 |
89.67 |
|
W |
9546 |
0.23 |
166.41 |
2.85 |
3.28 |
19.80 |
749.20 |
|
Y |
9546 |
1.91 |
45.60 |
7.13 |
8.46 |
1.28 |
0.31 |
|
Zn |
9546 |
5.87 |
1667.00 |
58.65 |
39.60 |
12.07 |
344.36 |
|
Zr |
9546 |
15.84 |
1724.36 |
200.32 |
89.32 |
2.69 |
20.76 |
|
Al2O3 |
9546 |
0.40 |
19.67 |
11.24 |
2.13 |
-0.71 |
0.84 |
|
CaO |
9546 |
0.11 |
47.46 |
3.09 |
4.42 |
3.31 |
14.76 |
|
Fe2O3 |
9546 |
0.27 |
39.77 |
2.96 |
1.61 |
5.97 |
114.58 |
|
K2O |
9546 |
0.11 |
7.98 |
3.18 |
0.85 |
0.24 |
0.46 |
|
MgO |
9546 |
0.04 |
27.58 |
1.07 |
1.04 |
7.60 |
109.50 |
|
Na2O |
9546 |
0.07 |
7.42 |
1.88 |
0.73 |
0.35 |
0.25 |
|
SiO2 |
9546 |
7.15 |
87.99 |
71.47 |
9.17 |
-2.05 |
6.55 |
|
Note: The units of Ag, Au, Cd, and Hg are ppb and the units of other elements are ppm, apart from oxides (wt.%). |
|||||||
(5) At many places, sentences are too long, which is frequently used, and may bring significant changes in the meanings of actual scientific investigations that has been discussed in this article. It should be re-phrased into small and meaning full sentences. That may be more appropriate in terms of technical documentation.
Thank you very much for your very meaningful suggestions. We have shortened and simplified complex sentences in accordance with technical requirements for research paper writing, aiming to make the content accessible and easy to understand. Additionally, we have rephrased the majority of expressions into concise and meaningful sentences.
(6) At several places, grammatical corrections are need? It is marked with yellow colour.
Thank you very much for your very constructive comments. We've made grammatical corrections to the yellow color markers in this manuscript.
For example:In the eastern section of the Gangdese metallogenic belt, which is known as the largest district for dense porphyry copper deposits, numerous large and super large deposits like Qulong, Tinggong, Chongjiang, Jiama, and Bairong have been discovered. However, in the western section of the belt, the number of discovered deposits remains relatively limited.
This study specifically concentrates on porphyry copper deposits in the western section of the Gangdese metallogenic belt and aims to address the limitations of the ilr-RPCA-back-clr model. To investigate the spatial distribution pattern of geochemical elements and identify real anomalies related to porphyry copper deposits, the study employs overlay analysis of local singular elements of geochemical combination elements.

Round 2
Reviewer 2 Report
My revision suggestions for the manuscript were taken into consideration by the authors. Thank you and I wish you success in your future work.
Reviewer 3 Report
Author has addressed all the comments and suggestions so I have no objection to accept themanuscript.
Much Better